# IL-33 and IL-37: A Possible Axis in Skin and Allergic Diseases

**DOI:** 10.3390/ijms24010372

**Published:** 2022-12-26

**Authors:** Francesco Borgia, Paolo Custurone, Federica Li Pomi, Mario Vaccaro, Clara Alessandrello, Sebastiano Gangemi

**Affiliations:** 1Department of Clinical and Experimental Medicine, Section of Dermatology, University of Messina, 98125 Messina, Italy; 2Department of Clinical and Experimental Medicine, School and Operative Unit of Allergy and Clinical Immunology, University of Messina, 98125 Messina, Italy

**Keywords:** IL-37, IL-33, skin, allergic diseases, inflammation, interleukin, atopic dermatitis, psoriasis, asthma, contact dermatitis

## Abstract

Interleukin (IL)-37 and IL-33 are among the latest cytokines identified, playing a role in several inflammatory conditions, spanning from systemic conditions to tumors to localized diseases. As newly discovered interleukins, their role is still scarcely understood, but their potential role as therapeutic targets or disease activity markers suggests the need to reorganize the current data for a better interpretation. The aim of this review is to collect and organize data produced by several studies to create a complete picture. The research was conducted on the PubMed database, and the resulting articles were sorted by title, abstract, English language, and content. Several studies have been assessed, mostly related to atopic dermatitis and immunologic pathways. Collective data demonstrates a pro-inflammatory role of IL-33 and an anti-inflammatory one for IL-37, possibly related to each other in an IL-33/IL-37 axis. Although further studies are needed to assess the safety and plausibility of targeting these two interleukins for patients affected by skin conditions, the early results indicate that both IL-33 and IL-37 represent markers of disease activity.

## 1. Introduction

As the current knowledge keeps expanding in the vast universe represented by skin and allergic conditions, cytokine networks represent one of the latest fields of research that lead to a better understanding of these diseases. The use of cytokines as therapeutic targets or activity markers triggered the interest of researchers to better define the cytokine activity and connection that lead to both immunological and clinical consequences of such interaction. Two of the latest interleukins ever discovered, IL-33 and IL-37, could be connected by a potential axis, featured in many skin and allergic conditions, and representing a missing puzzle piece in the complex picture of chronic skin and respiratory conditions.

### 1.1. Interleukins: The Proteinic Network behind Immune Responses and Therapies in Generalized Inflammatory States

The interleukins are a class of cytokines released both actively and passively by animal leukocytes as a means to coordinate the immune response. Originally thought to be secreted only by leukocytes, over the years, many other cell populations have been identified as interleukin secretors (macrophages, fibroblasts, endothelial cells, etc.). Their presence in the immune response is pivotal for many inflammatory processes, and the lack or over-production of such mediums leads to burdensome conditions [1,2,3]. The classification of interleukins has proven to be harder and harder as scientific progress proceeds, as each interleukin plays a different role in different conditions such as blood concentration, the presence of different receptor isoforms, the presence of antagonist and target cells. Roughly, they have been divided into pro-inflammatory and anti-inflammatory: to the first group belongs IL-1 and its related family, and IL-6; to the anti-inflammatory group belongs the antagonists of the pro-inflammatory ones, such as IL-10 and IL-11 [4]. The primary stimulus to produce a specific interleukin can be inscribed in a bigger picture involving other cells and the cytokines that they produce, but most of the time is represented by factors released upon cell death or damage [5]. These mediators, by linking to specific receptors on the surface of various immunity cells, lead to intracellular cascades that activate genes related to the immune response. These genes subsequently lead to the expression of specific proteins that lead to the furthering of such inflammatory status, for example, the nitric oxide synthase, receptors for tumor recognition, or other interleukin production/interaction [6,7]. The last finding is what led to the idea that interleukins interact intimately with each other by forming a network of information between cell populations, sometimes with a domino effect which can be resumed in the idea of interleukin axis. Such interactions have proven to be crucial in the pathogenesis of several chronic diseases and, by targeting them, useful for the clinical management of systemic conditions. By focusing on the relevance of interleukin axes in skin diseases, we can mention the IL-17/-23 axis, which is involved in psoriasis, the IL-4/-13 involved in atopic dermatitis, and IL-31/-33 in allergic diseases [8,9,10]. The IL-31/-33 axis is one of the latest axes to be identified, in which the effects of the two cytokines combined are greater than the mere sum of their single effects in perpetrating the inflammatory response of the host’s organism. However, this is not always the case, as other inflammatory axes can be involved in other diseases in which one interleukin plays as an inflammatory factor while the other one tries to contain such processes. One of these new potential axes could be represented by the IL-33/-37 axis.

### 1.2. IL-37

IL-37, a cytokine expressed in various inflammatory conditions, is one of the latest molecules identified in the IL-1 family [11]. IL-37 is present in human organisms in five isoforms, encoded as IL-37 from A to E, which result from a process of alternative splicing. IL-37b, which is the most common and heaviest of the five, is expressed by various cell populations, such as keratinocytes, monocytes, and immune cells [12]. This cytokine has been identified not just as an inflammation mediator but also as a transcriptional regulator, when incorporated inside the nucleus, thanks to its DNA-binding activity [2]. As part of the IL-1 family, this cytokine is mainly thought to play a role in innate immunity and can bind several receptors. First discovered in the early 2000s by several research groups, IL-37 (formerly known as IL-1F7) was found to be linked to another important cytokine involved in the inflammatory process: IL-18, and its soluble receptor, IL-18 binding protein (IL-18BP). IL-18, which acts as a promoter to interferon (IFN)-γ production, is inactivated by circulating levels of IL-18BP, but a study on psoriasis has shown that, in order to be deactivated, medium doses of IL-18BP need to be administered, while a higher dose does not produce the supposed anti-inflammatory effect [13]. This finding led to the conclusion that an orphan ligand might have been at play, which was later identified as IL-37. In fact, IL-37 can bind the receptor of IL-18, represented by IL-18 receptor alpha and, at the same time, the BP form. After its discovery, IL-37 was better characterized to understand its function, structure, and physiology. Encoded from chromosomes 2, 9, and 11 (9 of the total 11 genes are encoded from chromosome 2), this cytokine is present in five isoforms, whose most studied, common, and heavy form is IL-37b (218 amino acids). As for the other cytokines of the IL-1 family, IL-37 is synthesized as an inactive form that lacks the classic hydrophobic leader sequence and is then cleaved by caspase-1, although other enzymes may have a role in this process [4]. Subsequent studies found that the various isoforms of IL-37 are sometimes expressed in specific tissues in only one form, such as the brain (IL-37a) or heart (IL-37c), but further studies are required to explain the reason why these five isoforms are expressed selectively [14]. As for the receptor, IL-37 binds the receptor for IL-18 but with low affinity, not preventing the IL-18 binding. Researchers suggested that IL-37 might not just be a mere antagonist of the effects of IL-18 but rather, by binding to this receptor, might produce other effects. The alpha chain of the receptor, in fact, instead of pairing with the beta chain, might create dimers with another accessory chain which produces negative effects on the inflammatory environment. This explanation, moreover, would suggest why there are several isoforms of this cytokine: depending on the tissue and complementary chain of the alpha fragment of the receptor could explain different effects played in various body parts. As for its role, studies are starting to show an overall inflammation-dampening effect: a cell model demonstrated that cells transfected with IL-37, when put in lipopolysaccharide (LPS) rich medium [15], released less proinflammatory cytokines such as tumor necrosis factor (TNF)-α and IL-6. Independently of its role, levels of IL-37 have been assessed in several diseases, with higher levels in sera of patients affected by tumors, metabolic diseases, and cutaneous conditions [12].

### 1.3. IL-33

IL-33 is a 30 kDa cytokine that belongs to the IL-1 family and was previously known as IL-1F11. This interleukin binds to its receptor, ST2, which belongs to the toll-like receptor (TLR)/IL-1-receptor (IL-1R) superfamily. The ST2 receptor creates heterodimers together with the IL-1 receptor’s accessory protein (IL-1RAcP). The discovery of IL-33 allowed researchers to find the ligand for the former orphan receptor ST2, previously called IL-IR4, which is mainly linked to the Th-2 immune response [16]. The IL-33, in its premature form, is cleaved by a caspase in its final form, which is released after cellular damage. When cell injury or necrosis occurs, released IL-33 also acts as an alarmin, inducing M2 macrophages homing toward the site of inflammation and interleukin production in the surrounding cellular population, including mast cells, dendritic cells (DCs), and T cells. As for DCs, IL-33 induces their activation, driving the production of Th2-associated cytokines [17]. Moreover, IL-33 regulates innate immune cell activity, such as monocytes and tissue mast cells (MCs), via the activation of CD34+ cells and the recruitment of eosinophils. Given that the effects of this interleukin play a role in several lines of cells’ activation, IL-33 is involved in many different pathological conditions, such as cardiovascular diseases, arthritis, infective diseases, neurological conditions, allergic diseases, cutaneous conditions, and cancer, acting together with the neuropeptide substance P in inducing TNF-α in cultured MCs and enhancing the gene expression of mitogen-activated protein kinase (MAPK) and nuclear factor kappa-light-chain-enhancer of activated B cells (NF-κB), well-known factors involved in inflammatory pathways [16]. The research has preventively taken an interest in MCs as these cells are involved in the maturation of IL-33 from the immature form and are implicated in immunity and inflammation tolerance [18]. It is interesting that chymase, an enzyme stored inside MCs, can inactivate IL-33 as a defensive response to inflammation. Along with the lower activity of IL-33, studies on infective conditions found that IL-37 levels rise as lower levels of IL-33 are achieved as a means to lower local and systemic levels of inflammation. Given the double role of these interleukins and their common connection to other IL-1 family cytokines, it would be worth researching if there is a possible axis connecting these elements in response to early and late insults. Thus, the aim of this review is:Investigate the role of IL-33 and IL-37 and find possible connections between them.Focus on the role of IL-37 in cutaneous diseases and possible correlations with IL-33.Possible therapeutic usage of IL-37.

## 2. Materials and Methods

The research was carried out on the PubMed database, inserting the keywords “IL-33” and “IL-37”, focusing on dermatological and allergological articles. The preliminary research excluded previous reviews and systematic reviews, along with articles not in the English language. The results were screened and selected in the following order: title, abstract, and content. Double results were screened and removed from the final article count. Table 1 reports the articles that were included and reviewed.

## 3. Results and Discussion

### 3.1. The Axis IL-33/IL-37: What We Do Know

In contrast to the IL-31/IL-33 axis, one of the latest axis identified in inflammatory skin and allergic conditions, where actors act synergistically towards the inflammation pathway and stimulate each other by amplifying the inflammatory process, this new IL-33/IL-37 axis would appear to behave like a two-plate balance, where the imbalance towards one or the other side occurs because of partially unknown reasons [10,38]. While IL-33 plays just a pro-inflammatory role together with IL-31, such as in the case of its effects on innate lymphoid cells 2 (ILC2), IL-37 counterbalance the effects of IL-33 [37,39]. IL-33, paired with IL-31, has been studied in the ILC2 population, and recent evidence suggests both an anti-tumoral and a pro-tumorigenic role [40], as much as a worsening factor in oxidative stress environments such as some kidney diseases [41]. Taken alone, this evidence suggests that IL-33 plays just a pro-inflammatory role, while IL-31 enhances this process by furthering inflammation. It has been demonstrated, though, that after successful treatments, levels of IL-33 and IL-37 change together, possibly due to lower levels of cellular damage and a lesser need to contain local and systemic remodeling processes [42]. IL-37 acts mainly as an anti-inflammatory cytokine, whose effects are played primarily by the MCs [43], even though other cell populations resent its effects, such as endotheliocytes, keratinocytes, macrophages, and lymphocytes [44]. IL-37 produces its anti-inflammatory effects by reducing the intracellular mechanisms that lead to the activation of p38, extracellular signal-regulated kinase (ERK), and signal transducer and activator of transcription 1 (STAT1), which are pro-inflammatory factors involved in immune responses. It is arguable that, since IL-33 is regulated thanks to factors such as ERK, p38, and STAT1, IL-37 could also downregulate IL-33 by stopping the p38 pathway, as demonstrated by the fact that a knockdown of IL-37 induces the phosphorylation of p38 and c-Jun N-terminal kinases (JNK). Previous studies demonstrated that the phosphorylation of ERK and p38 is involved in the expression of IL-33 in keratinocytes, suggesting that the suppression of p38 by IL-37 leads to the indirect suppression of IL-33 [34]. Since IL-37 inhibits IL-33, and MCs produce IL-33, it is reasonable to study the inhibitory effect of IL-37 in MCs. IL-37 binding IL-18 receptor alpha chain acts as an inhibiting stimulus of inflammatory mediators, including TNF, IL-1, IL-6, IL-33, and nitric oxide (NO). The same effects have been registered in patients affected by adult-onset Still’s disease, where along with pro-inflammatory cytokines, the levels of IL-37 were also higher [45]. Innate immune cells such as DCs orchestrate inflammatory, autoimmune, and allergic responses, and in stimulated DCs, IL-37 inhibits Granulocyte-Macrophage Colony-Stimulating Factor (GM-CSF) and Macrophage Colony-Stimulating Factor (M-CSF) production and reduces the inflammatory response [46]. IL-37 also down-regulates chemokines such as Monocyte Chemotactic Protein (MCP)-5/Chemokine C-C motif ligand 12 (CCL12) and reduces CXC chemokines such as IL-8. IL-37 may interfere with the TLR4 signaling pathway, which is also involved in the innate immune system. Thus, IL-10 and IL-37 could serve as therapeutic targets for the treatment of the early stages of inflammatory disorders, as MCs are found around blood vessels and represent one of the first lines of defense for the host organism [16]. Another study suggests that the nuclear translocation of IL-37 is crucial for its anti-inflammatory function in vitro and in vivo. A failure in the cleavage process of the intracellular IL-37 precursor prevents the translocation of IL-37 into the nucleus, leading to a loss of the cytokine-inhibiting functions, although the suppression of innate inflammation is still maintained by surface receptor activity [25]. Moving to other diseases, in multiple sclerosis (MS), elevated levels of IL-37 correlate with disease severity as a means to mitigate the inflammation, as already noticed in other inflammatory conditions such as ankylosing spondylitis, rheumatoid arthritis, and asthma. IL-37 inhibits both the NF-κB signaling pathway and IL-1R8 and IL-18Rα receptors, de facto slowing down the pathogenetic processes that lead to disease manifestations. Moreover, there is a correlation between the extent of disability in MS and levels of IL-33, IL-37, and Vascular-Endothelial Growth Factor Receptor 2 (VEGFR2), and it has been shown that IL-33 might be a strong predictor factor of severe forms of MS independently, independently by sex, age, number of recurrences and disease duration [23]. The same consideration can be made for arthropathies, where the chronic inflammation component seems to trump the effect of IL-37 by either targeting the agonists of this process, such as IL-36 or strengthening the action of its antagonists [47]. In addition, in infectious diseases, the role of IL-33 and IL-37 has been evaluated. IL-33 acts as an alarmin and induces pro-inflammatory responses when released by traumatized or infected cells. On the contrary, IL-37 dampens immune responses, downregulating the reactions in macrophages to bacterial compounds such as the LPS [20]. The dual role of this correlation cytokines has been studied in the case of acute clinical conditions of paracoccidioidomycosis, in which IL-33 is released by damaged cells and induces the Th2 polarization of the immune response, the production of immunoglobulin E (IgE), and recruitment of eosinophils. IL-37, produced by macrophages, on the contrary, plays an important role in the suppression and regulation of this immune response, reducing the production of other pro-inflammatory cytokines without altering the production of cytokines with anti-inflammatory activity [24]. Another infection in which both these cytokines were studied is COVID-19. It has been demonstrated the worsening role of IL-33 and the protective features of IL-37: IL-33 stimulates cytotoxic T lymphocytes (CTL) activity and antibodies’ production, thus representing an important stimulus for the host’s defense while, on the contrary, impaired functions of IL-33 favor the viral infection tolerance. On the other hand, IL-37 administration ameliorates respiratory inflammation in COVID-19-infected mice, as low IL-37 levels, along with high IL-8 and C-reactive protein levels, predict a worse clinical outcome [26]. As a disease with a high social impact as COVID-19, a possible role of IL-37 could be interesting to study, but data are too scarce to draw any definitive conclusion.

### 3.2. Involvement of IL-33 and IL-37 in Clinical Conditions

#### 3.2.1. Atopic Dermatitis and Skin Irritation

IL-37 downregulates systemic and local inflammation, and its levels have been found to be higher in systemic inflammatory conditions. This finding is just apparently in contrast with another study, which correlated local levels of IL-37 and filaggrin expression in an atopic dermatitis (AD) model. The authors’ hypothesis, in fact, is that in order to enhance IL-37 production, keratinocytes should be more differentiated as the concentration of this interleukin increases as the most superficial layers of the epidermis are reached [48]. It can be suggested that, in fact, local levels may increase initially in a widely spread condition, while local levels decrease as the local damage piles up. Two of the main functions carried out by IL-37 revolve around the T regulatory cells (Treg) via expression of IL-10 and thus enhance their immunosuppressive function, IL-37 and local regulation of mammalian target of rapamycin (mTOR) signaling pathway, which acts as a regulator of autophagy [48]. Autophagy is one of the key players in infectious diseases, and its link to IL-37 has already been assessed in other conditions [49] and, as a common link between microbes and Tregs, IL-37 suppresses the infiltration of eosinophils and regulates the gut environment [33]. Another interesting field of research involves IL-37 and the senescence of the skin linked to inflammation. Some authors proposed the use of tapinarof, an enhancer of nuclear factor E2-related factor 2 (NRF2), whose levels of expression rise during tapiranof-based therapy, along with those of IL-37, thus reducing levels of inflammation in the skin [34]. Switching to IL-33, the Th2 response is responsible for the acute phase of AD and IL33, activating both MCs and the innate lymphoid cells (ILCs), leading to itching by the release of neural mediators [30,34]. IL-33 knockout mice do not present allergic inflammatory reactions, showing that this cytokine is crucial for the activity of AD [50], while mice lacking sebum present a dysregulation of skin flora and higher levels of IL-33 in the skin [51]. On the contrary, IL-37 executes its anti-inflammatory role by blocking the activation of pro-inflammatory signaling mediators, including p38, ERK, and STAT1. In fact, the knockdown IL-37 expression induces the phosphorylation of JNK, with a subsequent increase of the levels of IL-33 [34]. AD, presenting a complex pathogenesis, shows that several genetic factors are at play when IL-37 levels increase. Leading further studies in this direction, especially with “classic” therapies, which are already available worldwide, could explain the function of both these cytokines [52].

#### 3.2.2. Allergic Contact Dermatitis

IL-37 alleviates inflammation involved in contact dermatitis, such as swelling, neutrophil infiltration, and inflammatory cytokine production (TNF-α, IL-1β, IFN-γ, and IL-13). The recruitment of local MCs, along with the enhanced production of IL-33 in the ear dermis, on the other hand, leads to the amplification of pro-inflammatory cytokines, such as IL-6, TNF-α, IL-13, and MCP-1. In the same experimental model, increased levels of IgE were detectable in rats with dermatitis due to the increase in MC levels. As an interleukin directly related to MCs, IL-37 could act as an inhibitor of MC activation, including inhibition of IL-33 and suppressor of MC degranulation. This study also showed the involvement of NF-κB and P38 MAPK signaling in IL-33-induced MC inflammation. Since IL-37 has little effect on ST2 and Smad3 expression, this finding suggests that IL-33, being a cytokine with Th2 polarization function in certain conditions, may play the leading role in contact dermatitis [36].

#### 3.2.3. Psoriasis

Several studies demonstrate the role of IL-33 and IL-37 in the regulation of skin inflammation in psoriasis. IL-33, released by damaged keratinocytes, enhances the transcription of genes encoding psoriasis-related mediators in keratinocytes [34], while the psoriatic lesions express higher levels of IL-37 than the skin of healthy individuals [29]. In a study conducted in 2018, cytokine levels were studied in psoriasis patients, aiming to find a relationship between mediators and disease prognosis. The authors found a significant association between IL-33 and IL-36 serum levels with disease severity. There was no significant difference between gene expressions of IL-37 depending on the disease grading, but serum levels and IL-37 gene expression were significantly higher in psoriatic patients when compared to those in healthy controls. This finding of higher sera levels of IL-37 is in contrast with another study, which found that skin samples from psoriatic skin produced less IL-37, probably due to the thinning of the stratum granulosum [53]. Considering the anti-inflammatory effects of IL-37, increased serum levels of IL-37 in psoriatic patients should be considered as a compensatory response not yet enough to alleviate the inflammatory state of the disease [29]. This is an interesting finding, considering that one of the main secreting cells of this interleukin are MCs, a cell group intimately linked to blood vessels. With the secretion of IL-37, MCs inhibit the function of NF-κB and mitogen-activated protein kinase, thus suppressing early signs of inflammation, suggesting that IL-37 might not only play an important role in later stages of the disease but also during the first pathogenetic processes [54]. In addition, since a correlation between serum levels of IL-37 and psoriasis severity has been demonstrated, it seems that IL-37 may compensate for the inflammatory effects of IL-33, maybe by suppressing potential MAPK and STAT1 activation in human keratinocytes, in order to improve the skin function of the patients and the prognosis of the disease [34]. Finally, another interesting aspect to consider is the role of IL-37 in post-inflammatory hyperpigmentation: in fact, IL-37 seems to activate genes involved in melanogenesis, such as melanocyte-inducing transcription factor (MITF), explaining why, once the inflammation processes fade away and levels of IL-37 locally increase, discoloration patches substitute the previously psoriasis-affected areas [55]. Considering that both IL-33 and IL-37 could play a prominent role in psoriasis and, potentially, in its arthritic form, it is worth considering that targeting these cytokines could represent an option as a therapeutic target in the future [56].

#### 3.2.4. Melanoma

The role of IL-33 in the response of the host’s body to cancer cells has been evaluated in several forms of tumors [57,58,59], although the skin the studies are not quite enough to draw final conclusions, but perspectives. Perhaps, the most characterized of the skin tumors is represented by melanoma. As a pro-inflammatory cytokine, IL-33 represents one of the first actors at play in the modulation of the tumoral micro-environment, thanks to the activation of lymphoid cells and CD8+ [40]. This effect might be due to the interaction between IL-33 and ST2: The inflammation and toxicity generated by targeting and disrupting tumors could be inhibited by IL-37, leading to an improvement in the quality of life of patients. The expression of IL-37 shows upregulation of mRNA in tumors, also comprising melanoma. This enhancement occurs mainly in Treg cells and in granulocytes, also including MCs. The up-regulation of IL-37 RNA in tumors indicates a defensive state that the organism uses when affected by neoplasms [31]. Acting indirectly on the axis of ST2/IL-33, which appears crucial in Treg activity in the case of other forms of cancer [60], exploiting the beneficial effects of IL-37 upregulation could lead to lessening spreading processes while lowering local levels of inflammation.

#### 3.2.5. Asthma

IL-33 has been well assessed as a Th2-response promoter in allergic diseases: this cytokine, for example, in conjunction with airborne allergens, leads to sensitization and allergic inflammation due to *Alternaria* [61,62]. IL-33 promotes Th2 cell differentiation, ILCs activation, and production of IL-5 and IL-13 and activates airway epithelial cells by amplifying their release of IL-8. This process results in the upregulation of pro-inflammatory molecules (GM-CSF, ICAM1), while Th2-type cytokine levels in airway eosinophils and mucus production were considerably diminished in IL-33-deficient animals. All these pro-inflammatory effects deriving from IL-33 activity are regulated and diminished by IL-37, suggesting that this cytokine acts as a central factor for the maintenance of local immune homeostasis. Specifically, targeting these pro-inflammatory cytokines might be a better treatment than glucocorticoid drugs [37]. In addition, IL-37 might inhibit thymic stromal lymphopoietin (TSLP) production by the bronchial epithelial cells and, according to a study, might have the same effect as blocking the TSLP action [35]. The main pathways involving IL-37 and IL-33 are represented in Figure 1.

### 3.3. Potential Therapeutic Strategies of IL-33 and IL-37 Axis

As demonstrated by the collection of articles that deal with these two cytokines and skin/allergic conditions, targeting either of these cytokines could produce positive results for patients affected by these conditions. The potential role of IL-33 has been evaluated in systemic sclerosis (SS): interactions between IL-33 and ST2 showed that leads to fibrosis in bleomycin-induced fibrosis in mice, and its blockade could lead to local amelioration of the disease [63]. On the other hand, though, therapeutic effects could be achieved otherwise: a review assessed the role of bleomycin in clinical practice in the treatment of several skin conditions. The effect on tumoral diseases was limited, but there is a rationale for using this treatment in the case of skin warts [64]. In fact, in contrast to other infectious diseases in which the pro-inflammatory component is present, as in the case of the herpes virus, viral warts do not show positive staining to this cytokine [65]. It would be worth researching if treatments that elevate local levels of inflammation, such as bleomycin and photodynamic therapy, could elevate local levels of IL-33 to promote healing processes, as both have proven to be both effective and without relevant sequelae [66]. Another aspect of particular interest is the axis IL-33/ST2, which has been studied in some forms of cancer. Sporadic studies show that targeting this cytokine could prove useful for aggressive forms of cancer such as Merkel cell carcinoma [67], as it appears that the overexpression of this axis is present in the tumoral environment. The role of IL-37 as a marker of disease activity and the organism’s response to the tumor has been demonstrated in melanoma, in which an upregulation of this cytokine proves that the host is responding to the tumoral invasiveness [31]. The same response to an inflammatory stimulus has also been proven in psoriasis and AD as proof that this anti-inflammatory role serves as containment of disease spreading and limitation to the production of IL-33 [29,33]. Dosage of these cytokines could prove useful in disease monitoring, although not all studies seem to prove that these cytokines increase based on disease severity [68]. Another field that could be interesting to explore is the involvement in infectious diseases: as in the case of COVID-19, IL-37 levels are lower in non-responding patients and relate to a worse prognosis [26]. This data, as in the case of such a widespread and socially relevant condition as a pandemic, could be useful for future monitoring and assessment of patients affected by contagious diseases. On the therapeutic side, the blockade of the production of IL-33 or the increase of IL-37 could be especially useful for chronic inflammatory diseases: an analog of IL-37, for example, could halt the spreading of such conditions, demonstrated by the experimental use of specific plant extracts, preliminarily [30]. Nonetheless, it is worth considering that increasing local or systemic levels of this interleukin could lead to indirect activation of IL-33 since the caspase-1 shares a cleaving activity on both IL-33 and IL-37, thus increasing inflammatory and fibrotic processes [69]. As for the biological drugs, targeting IL-33 could prove useful for skin conditions in which the chronic inflammatory processes represent a threat greater than the primal stimulus, such as in the case of AD [50]. Although these possible target therapies sound promising, thorough studies must be conducted to better understand their potential and, given that IL-37 leads to a premature shift to the Th2 immune response, could lead to the worsening of other concomitant conditions in which the Th2 response is predominant. Moreover, given the widely unknown effects of IL-37 and IL-33 on other physiological processes, such as skin remodeling and scar formation [70], experimental studies should be conducted to test effects on the skin microenvironment to test possible side effects such as infection risk and wound healing [70]. A way to avoid possible side effects could be represented by adult cell reprogramming, as just affected/perilesional cells could be instructed to produce more IL-37, thus producing local healing and, indirectly, reducing IL-33 levels [71].

### 3.4. Role of the IL-33/IL-37 Interaction in Autoimmune Diseases

As mentioned above, both these interleukins seem to be involved in several autoimmune diseases. Taken singularly, both these cytokines were studied as biomarkers, therapeutic targets, and pathogenetic factors, with interesting results, although it must be mentioned that studies dealing with both at the same time are too few. Beyond the already cited psoriasis and AD, the two most interesting and common diseases to investigate would be systemic lupus erythematosus (SLE) and Behçet’s disease (BD). SLE is characterized by the elevation of several cytokines, which explain the inflammatory nature of this condition, and levels of IL-37 were found to be higher in the case of renal damage and loss of the complement fraction, both of which represent markers of disease severeness [72]. Along the same line, also levels of IL-33 increase as complement factors are depleted from the blood flow [73], and its effects relate to skin manifestations during cutaneous lupus erythematosus [74]. Both these findings suggest that these cytokines could act as severeness predictors and markers of activity [75]. As for BD, the vasculitic manifestation strike, not just the skin compartment but also other bodily districts. In the cerebrospinal fluid, levels of pro-inflammatory cytokines are inversely correlated with serum levels of IL-37, according to a study [76], while IL-37-stimulated DCs prevent the production of IL-1 and TNF-α [77]. As studies are too few to propose a proper correlation between these two cytokines, experimental dosages after proper therapy could represent a response marker in future clinical practice [78]. Figure 2 summarizes the relationships between IL-33 and IL-37, in the balance between pro and anti-inflammatory factors.

## 4. Conclusions and Future Perspectives

IL-37 represents a new molecule in the great chapter of inflammation-related cytokines. Along with IL-33, its function has been demonstrated in different diseases and conditions. There is certainly a role in inflammation dampening of the IL-37 and -33 axis, especially in cutaneous and allergic diseases. Higher levels of these cytokines in affected patients demonstrate an effort of the host’s cells to contain damage and slow down inflammatory processes, which, in the long term, could lead to tissue remodeling and permanent damage. The dosage of such interleukins could prove useful to acknowledge the activity and severeness of various chronic conditions, such as atopic eczema and psoriasis, or demonstrate the response to systemic therapies to identify potential interactions between cytokines that still need further assessment. Inhibitors or monoclonal antibodies which target this axis could represent a valid perspective for future therapies, possibly along with already validated drugs, to achieve complete disease clearance and ameliorate the quality of life of affected patients. To recognize the potential therapeutic role of these interleukins, it is crucial to lead further studies which assess the proper pathways triggered or inhibited by them in an effort to produce safer and more complete treatments for patients that are affected by such burdensome diseases.

## Figures and Tables

**Figure 1 ijms-24-00372-f001:**
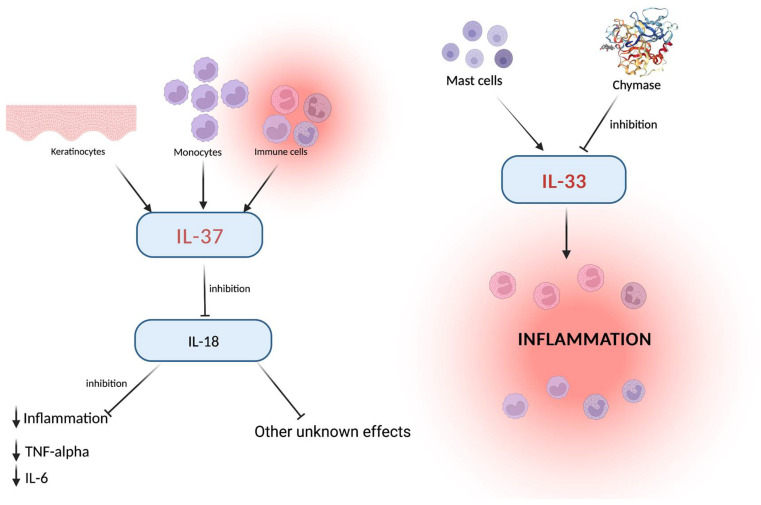
Keratinocytes, monocytes, and immune cells activate IL-37, which in turn downregulates IL-18 levels with consequent inhibition of inflammation and other effects which need further assessment. On the side, the IL-33 pro-inflammatory effect, played via the mast cells, explains the role of these cells around the blood vessels. The common link (the inflammation and related cells) represents the critical point in which other inflammation-related cytokines exert their role. Created with BioRender.com.

**Figure 2 ijms-24-00372-f002:**
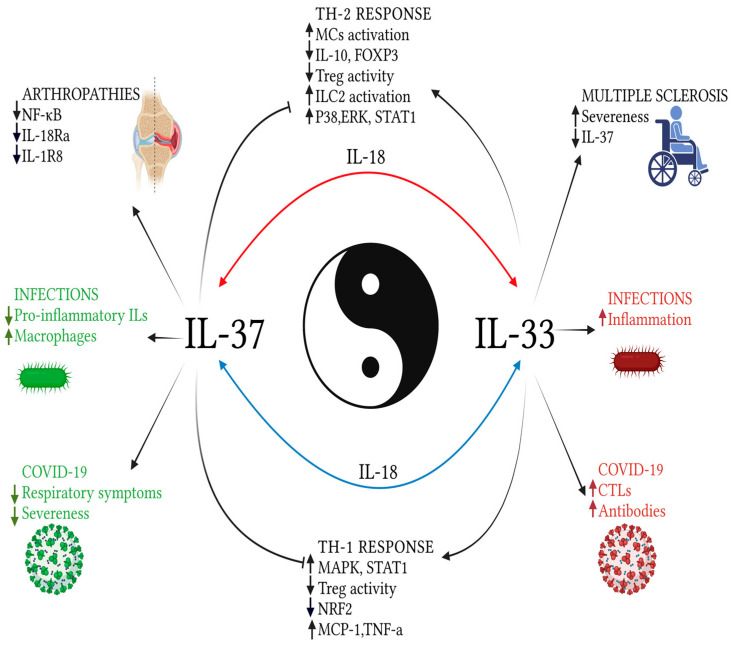
Explanatory image of the relationships between IL-33 and IL-37, in a balance between pro and anti-inflammatory factors. The effects on the Th-2 profile response are reported in the upper portion of the image; below, those towards the Th1 response. On the sides of the picture, the effects of the two examined cytokines on the response to emerging, infectious, and autoimmune diseases. While the effects on Th-1 and Th-2 responses have been well defined by some studies, those related to external pathologies have not evaluated the two cytokines together or require further studies, although they appear to have opposite roles, when taken separately. In the background, the mediator common to both cytokines: IL-18. Created with BioRender.com.

**Table 1 ijms-24-00372-t001:** Collection of articles included in the review and divided into three sections: IL-33/IL-37 axis, cutaneous diseases, and allergic diseases.

Author	Year	Cytokine or Axis	Disease	Summary
**IL-33/IL-37 Axis**				
Lopetuso et al. [19]	2013	IL-33 and IL-37	Gut diseases	IL-33 as a prototypic alarmin: - passively released upon cellular damage, stress, or necrosis - innate immune response signaling - physiological inflammation - gut homeostasis - a tool for tumor cells to create an ideal growth environment (IL-33/ST2 axis in tumor progression) IL-37 potential role in antibody production, B-cell activation, and colon tumorigenesis
Ross et al. [20]	2013	IL-33 and IL-37	-	IL-33 induces pro-inflammatory immune responses when released from destroyed cells IL-37 downregulates reactions in macrophages to limit the immune response
Li et al. [21]	2015	IL-37	Inflammation	IL-37 binds IL-18R for anti-inflammatory purposes
Montanari et al. [22]	2016	IL-33	Atherosclerosis	Endothelial cells as targets for IL-33 to exert its pro-inflammatory effects via GM-CSF and M-CSF production
Kouchaki et al. [23]	2017	Axis IL-33/IL-37	Multiple sclerosis	IL-33 as a predictor of severe forms of multiple sclerosis Levels of IL-37 correlate to disease severity and balance inflammation by inhibiting NF-κB signaling pathway and IL-1R8 and IL-18R α receptors
Alves et al. [24]	2018	Axis IL-33/IL-37	Paracoccidioidomycosis	- IL-33 release by cells - De novo production of IL-33 - Enhancement of Th2 cells and IgE production by B cells
Li et al. [25]	2019	IL-37	-	IL-37 suppresses innate inflammation
Makaremi et al. [26]	2022	IL-33 and IL-37	COVID	IL-33 can stimulate antiviral CTL activity and antibody production. Lower levels of IL-37 are associated with a higher risk of disease to COVID-19
Zhan et al. [27]	2021	IL-33 and IL-37	LES	Platelet count and IL-37 levels are correlated in patients affected by immune thrombocytopenia. IL-1β, IL-18, IL-36α, IL-36β, IL-36γ, IL-33, and IL-37 as biomarkers in the diagnosis of immune thrombocytopenia in LES
Conti et al. [16]	2017	IL-37	-	In stimulated DCs: - inhibits GM-CSF, M-CSF production - reduces inflammatory response in RA - downregulates CC chemokines and in neutrophils CXC chemokines - interfere with the TLR4 signaling pathway
**Cutaneous diseases**				
Kacem et al. [28]	2018	Axis IL-33/IL-37	Behcet skin lesions	- BD skin lesions are mediated by Th2 immunity (TSLP and IL-33) - Addition of recombinant IL-37 decreases expression of TSLP
Sehat et al. [29]	2018	IL-33 and IL-37	PSO	IL-33 levels correlate with the severity of the disease and IL-37 levels, a compensatory response
Tettamanti et al. [30]	2018	Axis IL-33/IL-37	AD	IL-33 activates MCs and ILC2, driving allergic inflammatory reactions in AD
Caraffa et al. [31]	2019	IL-37	Melanoma	Up-regulation of IL-37 RNA in tumors produces a defensive mechanism in the body of the host
Guttman-Yassky et al. [32]	2019	IL-33 and IL-37	AD	- IL-33 intense epidermal staining in both affected and unaffected AD skin - IL-37 downregulated in AD skin
Hou et al. [33]	2020	IL-37	AD	IL-37b suppresses innate immunity by reducing the infiltration of eosinophils and induction of Foxp3+ Treg cells
Tsuji et al. [34]	2022	Axis IL-33/IL-37	AD/PSO	- IL-37 downregulates IL-33 in keratinocytes: - IL-33 activates ILC2 and induces itching by stimulating nerves - IL-33 enhances the transcription of psoriasis-factors in keratinocytes in an autocrine mechanism - IL-37 reduces the activation of pro-inflammatory signaling mediators (p38, ERK, and STAT1)
**Allergic diseases**				
Berraïes A et al. [35]	2016		Asthma	- IL-33 induces TSLP production by epithelial cells of the airway tract - IL-37 suppresses TSLP production by bronchial epithelium cells
Li et al. [36]	2020	IL-37	DAC	- IL-37 alleviates inflammation in contact dermatitis via up-regulation of IκB expression and down-regulation of of NF-κB p65, NF-κB p65, and STAT3 - Smad3 is involved in the IL-37 cascade
Schröder et al. [37]	2022	IL-37	Asthma	IL-37 efficacy in dampening IL-33 effects is comparable to use of corticosteroids

## Data Availability

Not applicable.

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
