# Peer review of "IL-33 and IL-37: A Possible Axis in Skin and Allergic Diseases"

_ijms, 2022, doi:10.3390/ijms24010372_

Round 1
Reviewer 1 Report (Previous Reviewer 2)
No further comments
Author Response
Please see the attachment.

Reviewer 2 Report (New Reviewer)
Please see attachment

Author Response
Please see the attachment.

Reviewer 3 Report (New Reviewer)
The review is very interesting. For me is necessary to enlarge the chapter dedicated to the cytokines (and dedicate to an apposite chapter not only one introduction) starting from the principle that this kind of document may interest many readers and in this context is necessary to have a general idea before to focalize the attention on one or two.
Round 2
Reviewer 2 Report (New Reviewer)
The authors revised their manuscript in response to the suggestions. I have no other comments.
Reviewer 3 Report (New Reviewer)
The authors have answered correctly to my question.
This manuscript is a resubmission of an earlier submission. The following is a list of the peer review reports and author responses from that submission.
Round 1
Reviewer 1 Report
The overall concept of the ms is good but lacks novelty and general significance to meet the standard criteria of this prestigious journal. Most of the studies have been published previously in the same journal. For instance;
https://doi.org/10.3390/ijms20235856
Besides, plagiarism is suspected in most parts of the article. For instance
1) www.frontiersin.org : 5%
2) mdpi-res.com : 5%
Hence, the paper does not fulfill the standard criteria of the journal.
The abstract and the introduction section are below the standard.
The methodology is missing in the abstract section.
The provided literature is not sufficient to justify the author's claims.
Overall the paper is not scientifically robust and interesting for the reader.
Will not add a significant contribution to the field.
Reviewer 2 Report
This is an interesting review, I enjoyed reading. The authors however need to correct several grammatical points and provide a figure that would better support the main argument of the paper, the existence of an IL-33/IL-37 axis.
Page 3, line 114: “We carried out a Pubmed…” do you mean query? Research?
Page 6, line 120: “IL-33/IL-3” do you mean IL-33/IL-37 ?
Page 7, line 174: “antibodies productions” should be “antibody production”
Page 7, line 183: “Th2-oriented” should be “the Th2-oriented”; this whole paragraph is highly speculative
Page 7, line 212: “DAC” is not defined
Page 8, lines 239-240: “it has been demonstrated” : “it” should be deleted and “has been demonstrated” should go at the end of the sentence.
Page 8, line 263: “pro-asthmatic” do you mean “anti-asthmatic”? If IL-37 is anti-inflammatory why is it pro-asthmatic?
Page 9, figure 1: The figure doesn’t show any interaction between IL-37 and IL-33 to justify the existence of an IL-33/IL-37 axis which is the main point of the paper. The authors should provide a better figure that would show such an interaction, even if putative.
Page 9, lines 290-298: The sentence is too long. The authors should break it into smaller sentences.
Page 10: reference 7 is a duplicate of reference 4.